# Detection of pathogenic bacteria in ticks from Isiolo and Kwale counties of Kenya using metagenomics

Bryson Brian Kimemia[1,2]*, Lillian Musila[1,3], Solomon Langat[4], Erick Odoyo[1], Stephanie Cinkovich[5], David Abuom[1], Santos Yalwala[1], Samoel Khamadi[4], Jaree Johnson[6], Eric Garges[1], Elly Ojwang[1], Fredrick Eyase[1,4]

**1** Department of Emerging Infectious Diseases, United States Army Medical Research Directorate-Africa (USAMRD-A), Nairobi, Kenya, **2** Jomo Kenyatta University of Agriculture and Technology (JKUAT), Nairobi, Kenya, **3** Kenya Medical Research Institute (KEMRI), Centre for Microbiology Research, Nairobi, Kenya, **4** Kenya Medical Research Institute (KEMRI), Centre for Virus Research, Nairobi, Kenya, **5** United States Armed Forces Health Surveillance Division, Global Emerging Infections Surveillance Branch, Silver Spring, Maryland, United States of America, **6** United States Armed Forces Pest Management Board, Silver Spring, Maryland, United States of America

* brianbryson39@gmail.com

**Data Availability Statement:** The sequence data has been uploaded to NCBI and can be accessed under the BioProject Accession Number PRJNA1053293. The sequence data has been

## Abstract

Ticks are arachnid ectoparasites that rank second only to mosquitoes in the transmission of human diseases including bacteria responsible for anaplasmosis, ehrlichiosis, spotted fevers, and Lyme disease among other febrile illnesses. Due to the paucity of data on bacteria transmitted by ticks in Kenya, this study undertook a bacterial metagenomic-based characterization of ticks collected from Isiolo, a semi-arid pastoralist County in Eastern Kenya, and Kwale, a coastal County with a monsoon climate in the southern Kenyan border with Tanzania. A total of 2,918 ticks belonging to 3 genera and 10 species were pooled and screened in this study. Tick identification was confirmed through the sequencing of the Cytochrome C Oxidase Subunit 1 (*COI*) gene. Bacterial 16S rRNA gene PCR amplicons obtained from the above samples were sequenced using the MinION (Oxford Nanopore Technologies) platform. The resulting reads were demultiplexed in Porechop, followed by trimming and filtering in Trimmomatic before clustering using Qiime2-VSearch. A SILVA database pretrained naïve Bayes classifier was used to classify the Operational Taxonomic Units (OTUs) taxonomically. The bacteria of clinical interest detected in pooled tick assays were as follows: *Rickettsia spp.* 59.43% of pools, *Coxiella burnetii* 37.88%, *Proteus mirabilis* 5.08%, *Cutibacterium acnes* 6.08%, and *Corynebacterium ulcerans* 2.43%. These bacteria are responsible for spotted fevers, query fever (Q-fever), urinary tract infections, skin and soft tissue infections, eye infections, and diphtheria-like infections in humans, respectively. *P. mirabilis*, *C. acnes*, and *C. ulcerans* were detected only in Isiolo. Additionally, *COI* sequences allowed for the identification of *Rickettsia* and *Coxiella* species to strain levels in some of the pools. Diversity analysis revealed that the tick genera had high levels of Alpha diversity but the differences between the microbiomes of the three tick genera studied were not significant. The detection of *C. acnes*, commonly associated with human skin flora suggests that the ticks may have contact with humans potentially exposing them to bacterial

assigned BioSample accession numbers SAMN38855066 to SAMN38855120. The sequence data has been assigned SRA accession numbers (accession numbers - SRS19927750, SRS19927751, SRS19927763, SRS19927772, SRS19927785, SRS19927795, SRS19927801, SRS19927802, SRS19927803, SRS19927804, SRS19927752, SRS19927753, SRS19927754, SRS19927755, SRS19927756, SRS19927757, SRS19927758, SRS19927759, SRS19927760, SRS19927762, SRS19927761, SRS19927764, SRS19927765, SRS19927767, SRS19927766, SRS19927768, SRS19927769, SRS19927770, SRS19927773, SRS19927771, SRS19927774, SRS19927775, SRS19927776, SRS19927777, SRS19927778, SRS19927779, SRS19927780, SRS19927781, SRS19927782, SRS19927784, SRS19927783, SRS19927786, SRS19927787, SRS19927789, SRS19927788, SRS19927790, SRS19927792, SRS19927791, SRS19927793, SRS19927794, SRS19927798, SRS19927799, SRS19927797, SRS19927796, SRS19927800).

**Funding:** This work was funded by the Armed Forces Health Surveillance Branch (AFHSB) and its Global Emerging Infections Surveillance (GEIS) Section, FY2022 ProMIS ID: P0116_22_KY and FY2023 ProMIS ID P0094_23_KY.

**Competing interests:** The authors have declared that no competing interests exist.

infections. The findings in this study highlight the need for further investigation into the viability of these bacteria and the competency of ticks to transmit them. Clinicians in these high-risk areas also need to be appraised for them to include Rickettsial diseases and Q-fever as part of their differential diagnosis.

## Introduction

Interaction between humans and domestic animals creates a pathway for ectoparasites, such as ticks, leading to the emergence of zoonotic diseases—an issue of increasing concern according to the World Health Organization's One Health concept [1–4]. Among these diseases, tick-borne bacterial infections like Lyme disease, anaplasmosis, ehrlichiosis, and rickettsiosis pose significant global health threats [5]. These diseases not only cause debilitating symptoms but, if left untreated, can result in chronic health issues and fatalities, imposing a substantial burden on healthcare systems and compromising a country's economy [5,6].

Despite the global prevalence of tick-borne pathogens, accurate diagnostics are limited, potentially under-reporting their actual occurrence [7,8]. In regions like Kenya, with diverse tick fauna and abundant livestock and wildlife populations, Lyme disease (*Borrelia burgdorferi*), anaplasmosis (*Anaplasma phagocytophylum*), and rickettsial diseases have been reported with varying prevalence [9,10]. In the case of Lyme disease, only two cases have been reported in the country [11]. Most studies conducted in the country have focused on veterinary cases of tick-borne pathogens which leaves an incomplete picture when it comes to human disease. Spotted Fever Group of *Rickettsiae* such as Rocky Mountain Spotted Fever, Rickettsial-pox, and Boutonneuse fever have been reported in North and South America, Europe, Africa, and Asia. The most reported Spotted Fever Group of *Rickettsiae* in the United States is African Tick Bite Fever, caused by *R. africae* infection after international travel from Africa to the United States [12] Continuous surveillance and research are vital to understanding the distribution and impact of these diseases, crucial for developing effective prevention and control strategies [10].

This study addresses this gap by focusing on two specific regions in Kenya: Isiolo and Kwale counties. Isiolo is an arid/semi-arid county where livestock keeping constitutes the primary income source for approximately 80% of its population [13]. Hosting a large livestock market that attracts sellers from surrounding counties [14], it represents a critical hotspot for understanding zoonotic disease dynamics. Kwale, primarily an agricultural county, is home to a significant livestock-keeping community, especially in the Nyika plateau region [15]. Kwale County also borders Tanzania; this provides a cross-border importation route for vectors and by extension pathogens from the southern border. These unique economic activities and environmental characteristics make both Isiolo and Kwale ideal study areas, offering valuable insights into the prevalence and diversity of tick-borne bacterial diseases in distinct contexts.

By utilizing 16S rRNA gene sequencing, this research aims to comprehensively analyse the microbiomes of ticks collected from these regions. The study's objectives include assessing bacterial diversity through metagenomic analysis, the identification of potential and novel pathogenic bacteria, and confirming the taxonomic identity of tick species carrying tick-borne bacteria through metabarcoding. This focused approach ensures a nuanced understanding of the prevalence and nature of tick-borne diseases, essential for tailoring targeted public health management and prevention strategies in these areas.

## Materials and methods

### Ethical approval

The study was approved by the Kenya Medical Research Institute (KEMRI) Scientific and Ethics Review Unit (SERU) under protocol number KEMRI/SERU/CCR/4431. The study was also submitted to the National Commission for Science, Technology & Innovation (NACOSTI) and approved under license number 421618. The study was submitted for approval to the Walter Reed Army Institute of Research (WRAIR) Human Subjects Protection Branch (HSPB) and Institutional Review Board (IRB) under package WRAIR #3000. The study was exempted from requiring any further approval as per WRAIR policy #25 because no samples, data or information was being collected from human subjects. Consent was sought from farmers and pastoralists using an informed consent form.

### Sampling and pooling

Ticks were collected from cattle, sheep, goats, and camels from Isiolo and Kwale counties. The ticks were collected by a veterinarian through grooming. Sites from Isiolo County were Merti, Shambole, Isiolo market, and Isiolo slaughterhouse (Fig 1a). Isiolo had an average daytime temperature of 28˚C and a night temperature of 20˚C on the days sampling was done. This period, 9th June to 13th June 2022, also marked the end of the rainy season and the beginning of the dry season. These days had a relative humidity in the range of 69–88%. Two sites were sampled in Kwale County namely, Kisima and Mlalani (Fig 1b). Kwale had an average daytime temperature of 27˚C and a night temperature of 19˚C. The collection was done between 24th and 25th June 2022 in the rainy season. These days had a relative humidity in the range of 75–82%.

Sampled ticks were placed in 15 mL centrifuge tubes (Corning Inc, Corning, NY, USA) and transported under dry ice to the KEMRI/ United States Army Medical Research Directorate—Africa (USAMRD-A) laboratories. After the reception, the ticks were identified morphologically under a microscope using taxonomic keys [16–20] and pooled in groups of 1 to 8 individuals based on species, size, animal host, and site of collection before transferring to cryovials for storage at -81˚C.

### Surface sterilization and homogenization

To ensure the removal of external bacterial contaminants, surface sterilization was performed on the ticks. Briefly, cryovials were placed on ice in a biosafety cabinet, and 1 mL of 5% Sodium

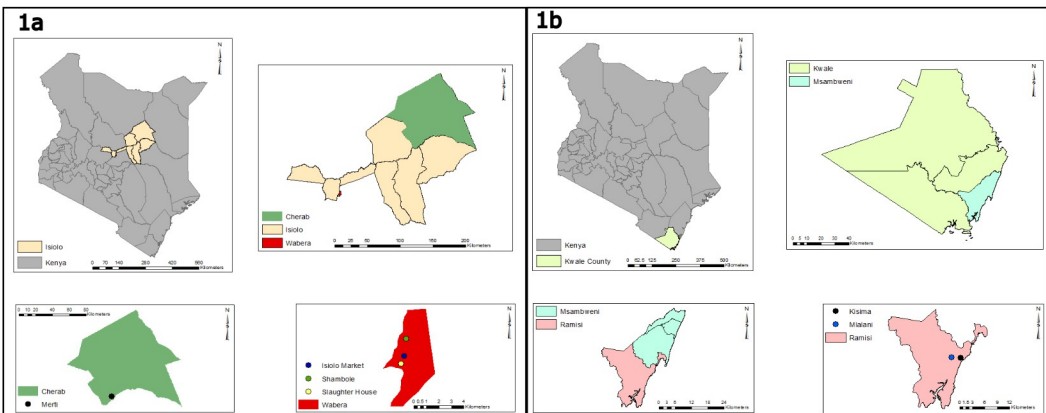

**Fig 1. Map of Sampling Sites, Isiolo County (a) Kwale County (b).** These maps were generated using ArcGIS version 10.2.2 for Desktop (Advanced License) courtesy of Samwel Owaka.

Hypochlorite solution was dispensed into the open cryovials and allowed a contact time of 5 mins. The ticks were then rinsed 3 times with distilled water.

Following surface sterilization, homogenization was carried out as follows: the ticks were frozen at -81˚C for a minimum of 30 minutes or exposed to liquid nitrogen in tubes with bashing beads to render them brittle. The ticks were then homogenized using an Omni bead Ruptor 24 (Omni International, Inc, Kennesaw, GA, USA) for two cycles, running at 4 m/s for 60 seconds with a 10-second pause between cycles. 1000 μL of homogenization media (40% glycerol, without antibiotics/antimycotic) was added followed by pulse vortexing to mix. The homogenate was then centrifuged at 10,000 rpm for 10 minutes. Subsequently, 200 μL of the supernatant was withdrawn for DNA extraction, while the remainder of the homogenate was stored at -81˚C for future use.

## DNA extraction and PCR amplification

DNA extraction was performed using the Quick-DNA™ Fungal/Bacterial Miniprep Kit (Zymo Research, Orange, CA) as per the manufacturer's instructions. Two separate PCR amplifications were performed to amplify the 16S rRNA and *COI* genes. The first was for bacterial metagenomics while the second was to confirm the morphological tick species identification.

For 16S rRNA PCR amplification, 10 μM of 27F (5′-AGAGTTTGATCCTGGCTCAG-3′) forward primer and 10 μM of 1492R (5′-TACGGYTACCTTGTTACGACTT-3′) [21] reverse primer pair used whereas for the Cytochrome Oxidase Subunit 1 (*COI*), 10μM of LCO1490 (5'-GGTCAACAAATCATAAAGATATTGG-3') forward primer and 10 μM of HC02198 (5'-TAAACTTCAGGGTGACCAAAAAATCA-3') [22] reverse primer pairs were used. The NEBNext High Fidelity 2x PCR Master Mix (New England Biolabs, Ipswich, MA) was used according to the manufacturer's instructions. The PCR cycles were as follows; For 16S rRNA, 95˚C initial denaturation for 5 min, 35 cycles of 95˚C denaturation for 30 sec, 53.8˚C annealing for 30 sec, and 72˚C extension for 1.5 min, and a final extension at 72˚C for 7 min. The *COI* PCR amplification cycle was as follows; 95˚C initial denaturation for 5 min, 35 cycles of 95˚C denaturation for 30 sec, 49˚C annealing for 30 sec, and 72˚C extension for 45 sec and a final extension at 72˚C for 7 min.

A 0.8%w/v TBE agarose gel was used to confirm the presence of bands at approximately 1500 base pairs (bps) size for the 16S rRNA gene and 710 bps for the *COI* gene, indicating successful amplification.

## Sequencing

After DNA extraction and PCR amplification, amplicons from both 16S rRNA and *COI* genes were quantified to ensure that they attained the concentration (15.58ng/μl) for the Native Barcoding Kit (SQK-NBD.112.96) (Oxford Nanopore Technologies, Oxford, United Kingdom). The sequencing was performed on a MinION MK1B (Oxford Nanopore Technologies, Oxford, United Kingdom) platform using an R9.4.1 flow cell (Oxford Nanopore Technologies, Oxford, United Kingdom). Data acquisition was done using the MikNOW Version 22.10.7 (Oxford Nanopore Technologies, Oxford, United Kingdom) software.

## Bioinformatics analysis

The raw sequencing reads (fast5 files) were base called using Guppy Version 6.3.8 (Oxford Nanopore Technologies, Oxford, United Kingdom) into FastQ files. Quality control checks were done using PycoQC Version 2.5.2 [23]. De-multiplexing, removal of barcodes, and adapters were done in Porechop Version 0.2.4 [24] after which Trimmomatic Version 0.36 [25] was used for quality filtering and trimming.

16S rRNA gene amplicon reads between 1,100 and 1,700 bps and *COI* reads between 600 and 850bps were selected.

16S rRNA amplicon reads were imported into QIIME2 [26] for de-replication, filtered for chimeras, and clustered in VSearch [27] and the Operational Taxonomic Units (OTUs) classified using a naïve Bayes classifier [28] trained on 16S rRNA data from the SILVA-138-99 database. Using controls included in the sequencing run, common laboratory contaminants present were filtered out and potential pathogenic bacteria were selected.

The *COI* reads were then submitted to the BugSeq metagenomic platform for the identification of both Vectors and bacteria that use oxidative phosphorylation during energy production. The taxonomic ID of the tick vectors was then confirmed on the MIDORI2 server [29] (VersionGB252) using the RDP classifier option [30].

## Data analysis

The OTU count data was first cleaned in Microsoft Excel by removing any blank spaces and confirming the alignments of columns and rows after which they were imported into RStudio (v 2023.06.1 Build 524). Alpha diversity was measured using the Shannon and Simpson indices per tick genus at the bacterial order, genus, and species level. The Shannon index measures the richness and diversity of OTUs in the genus while the Simpson index measures the Alpha diversity that provides information regarding the overall number of total individual OTUs. The indices were tested for distribution normalcy using the Shapiro-Wilk normalcy test to determine whether one-way ANOVA for normally distributed data or the Kruskal-Wallis non-parametric test was appropriate for statistical significance testing between the indices. Mann-Whitney U test was used to test the differences between the median number of reads for each order as a measure of abundance. This was followed by Dunn's multiple comparison tests between each of the orders.

Beta diversity between the tick genera was determined using PERMANOVA. The similarity between the number and types of OTUs between the genera was tested and a Bray-Curtis Dissimilarity matrix was generated. This matrix was then tested using 999 permutations of PERMANOVA. This allowed for model testing to determine the effect of each of the OTUs on the diversity of bacteria in all the ticks sampled at an isolated (ignoring other OTUs), individual (keeping other OTUs constant), and group (accounting for all OTUs) level.

The Minimum Infection Rates (MIR) per 100 ticks were calculated using the following formulae using a conversion factor of 100 [31].

$$MIR = \left( \frac{Number\ of\ Positive\ Pooled\ Samples}{Total\ Number\ of\ Ticks\ Tested} \right) \times Conversion\ Factor$$

This represents the minimum number of infected ticks in a group of 100 ticks.

The skewed sampling of the ticks between Isiolo and Kwale prevented any inter-county comparative analysis.

## Results

### Tick collection and identification

A total of 2,918 ticks were collected from both counties. Isiolo County contributed 2,858 ticks [Isiolo slaughterhouse(n = 1,592), Isiolo market(n = 1,019), Merti(n = 246), and Shambole (n = 1)]. Kwale County contributed 60 ticks [Mlalani(n = 36) and Kisima(n = 24)] (Table 1). A vast majority of the ticks (97.94%) were collected from Isiolo and the rest from (2.06%) Kwale. The ticks investigated in this study, belonged to 3 genera and 10 species (Fig 2): *Ambylomma*

**Table 1. Ticks collected from Isiolo and Kwale counties based on host and tick species.**

| Species | Cow | Sheep | Goat | Camel | Dragging | |
|---|---|---|---|---|---|---|
| *Hyalomma marginatum rufipes* | 32 | 0 | 0 | 0 | 0 | **Isiolo** |
| *Hyalomma truncatum* | 468 | 28 | 32 | 596 | 0 | |
| *Hyalomma dromedarii* | 95 | 6 | 44 | 167 | 0 | |
| *Hyalomma albiparmatum* | 17 | 6 | 1 | 2 | 0 | |
| *Amblyomma gemma* | 282 | 46 | 91 | 89 | 0 | |
| *Amblyomma lepidium* | 386 | 62 | 111 | 42 | 0 | |
| *Rhipicephalus pulchellus* | 56 | 16 | 25 | 93 | 1 | |
| *Rhipicephalus appendiculatus* | 1 | 1 | 2 | 0 | 0 | |
| *Amblyomma variegatum* | 0 | 0 | 1 | 0 | 0 | |
| *Rhipicephalus boophilus microplus* | 59 | 0 | 0 | 0 | 0 | |
| **Isiolo Sub-total** | **1396** | **165** | **307** | **989** | **1** | |
| *Amblyomma variegatum* | 4 | 0 | 1 | 0 | 0 | **Kwale** |
| *Rhipicephalus appendiculatus* | 23 | 0 | 18 | 0 | 0 | |
| *Rhipicephalus boophilus microplus* | 6 | 0 | 8 | 0 | 0 | |
| **Kwale Sub-Total** | **33** | **0** | **27** | **0** | **0** | |
| **Grand Total** | **1429** | **165** | **334** | **989** | **1** | **2918** |

*gemma* (17.4%), *Amblyomma lepidium* (20.6%), *Amblyomma variegatum* (0.2%), *Rhipicephalus appendiculatus* (1.5%), *Rhipicephalus boophilus microplus* (2.5%), *Rhipicephalus pulchellus* (6.5%), *Hyalomma dromedarii* (10.7%), *Hyalomma truncatum* (38.5%), *Hyalomma marginatum rufipes* (1.1%) and *Hyalomma albipamartum* (0.9%) collected during June 2022. *Rh. boophilus microplus* was initially identified as *Rh. boophilus decoloratus* during taxonomic classification based on morphology.

These ticks were then pooled based on tick species, animal host, and collection site resulting in 472 pools representing an average of 8 individual ticks per pool. Aliquots from the original pools were further combined into 69 super-pools representing an average of 50 ticks based on site and species.

**Confirmation of vector identity using Cytochrome Oxidase Subunit I sequences.** *COI* sequence analysis confirmed the identities of all tick vectors reliably to the Genus level,

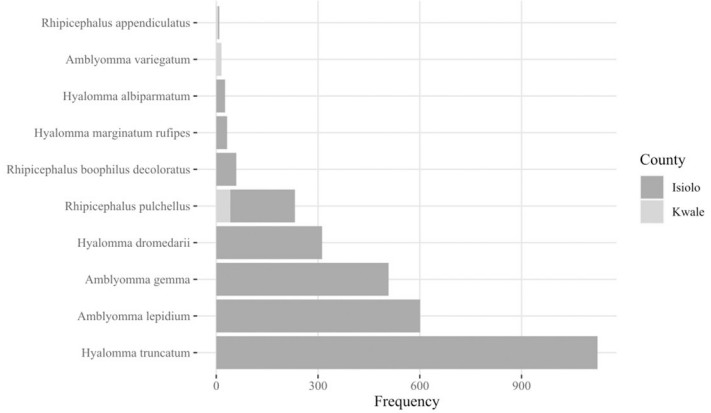

**Fig 2. Distribution of tick species collected from Isiolo and Kwale counties, individual species collected based on Cytochrome C Oxidase subunit I Identification.**

however, there were some discrepancies in 4.5% of the species between *COI* and morphological identification. Whereas these had been morphologically identified as *Rh. boophilus decoloratus COI* results identified them as *Rh. boophilus microplus*. The latter was used as the species identifier in this study.

## Sequencing output

The 16S rRNA gene sequencing run generated 694.31 million bases in 547,780 reads with an estimated N50 of 1.52kb while the *COI* gene sequencing run generated 381.11 million bases in 430,510 reads with an estimated N50 of 845kb. This achieved the target number of reads (10,000 reads) per sample.

## Bacterial diversity and richness in different tick genera

Alpha diversity was determined using the Shannon (a measure of richness) and Simpson (a measure of evenness) indices (Table 2). The Shapiro-Wilkins' test of normalcy revealed significant deviations from a normal distribution of the Shannon index at the order level ($p = 3.352e$-08), genus level ($p = 3.337e$-08), and species level ($p = 9.342e$-05). This was also the case for the Simpson index order level ($p = 2.774e$-10), genus level ($p = 9.923e$-10), and species level ($p = 1.482e$-05).

This informed the decision to use the Kruskal-Wallis test over the one-way ANOVA as the test for statistical significance between the indices at the different taxonomic levels. The results of the test showed no significant difference between the Shannon index and Simpson indices at the bacterial order, bacterial genus, and bacterial species levels ($p > 0.05$) when compared between the tick genera. This established that the microbiomes in the three tick genera had insignificant variations (Table 2). Comparative analysis between the two Alpha diversity indices showed that the level of richness reduces the lower the identified bacterial taxonomy level, however, this drop in the numbers of bacterial OTUs identified also seems to improve the distribution of the bacterial OUTs in the compared tick genera (Fig 3) which shows that a few of the OTUs had a big impact on diversity increasing the overall numbers of OTUs but not the general diverseness. Alpha diversity analysis highlighted the big impact the *Rickettsia* genus had on the number of bacterial units identified at the bacterial genus and species level. This was because the *Rickettsia* genus could not be speciated using 16S rRNA sequences.

**Table 2. The alpha diversity of identified bacterial read clusters at the bacterial order, genus, and species level.**

| Tick Genus | Mean Shannon index | Mean Simpson index | Mean Observed OTUs | Bacterial Taxonomic Level |
|---|---|---|---|---|
| *Amblyomma* | 1.227464 | 0.6415212 | 4.772727 | Order |
| *Hyalomma* | 1.153616 | 0.6144811 | 4.260870 | |
| *Rhipicephalus* | 1.100245 | 0.5935945 | 4.000000 | |
| *Amblyomma* | 0.9265109 | 0.5036514 | 3.590909 | Genus |
| *Hyalomma* | 1.0682956 | 0.5701056 | 4.086957 | |
| *Rhipicephalus* | 1.2765862 | 0.6632471 | 4.800000 | |
| *Amblyomma* | 0.5143822 | 0.2862695 | 2.454546 | Species |
| *Hyalomma* | 0.6816577 | 0.3794447 | 2.869565 | |
| *Rhipicephalus* | 0.8680247 | 0.4819174 | 3.400000 | |

The indices measured were Shannon (richness of unique clusters) where higher values indicate a higher number of unique bacterial reads identified at the respective level and Simpson (evenness of numbers in clusters identified) which ranges from 0(all unique clusters were identified at an equal number thus infinite evenness) to 1 (one of the unique cluster accounts for nearly all the identified clusters thus no evenness). The indices were measured according to the three tick genera sampled.

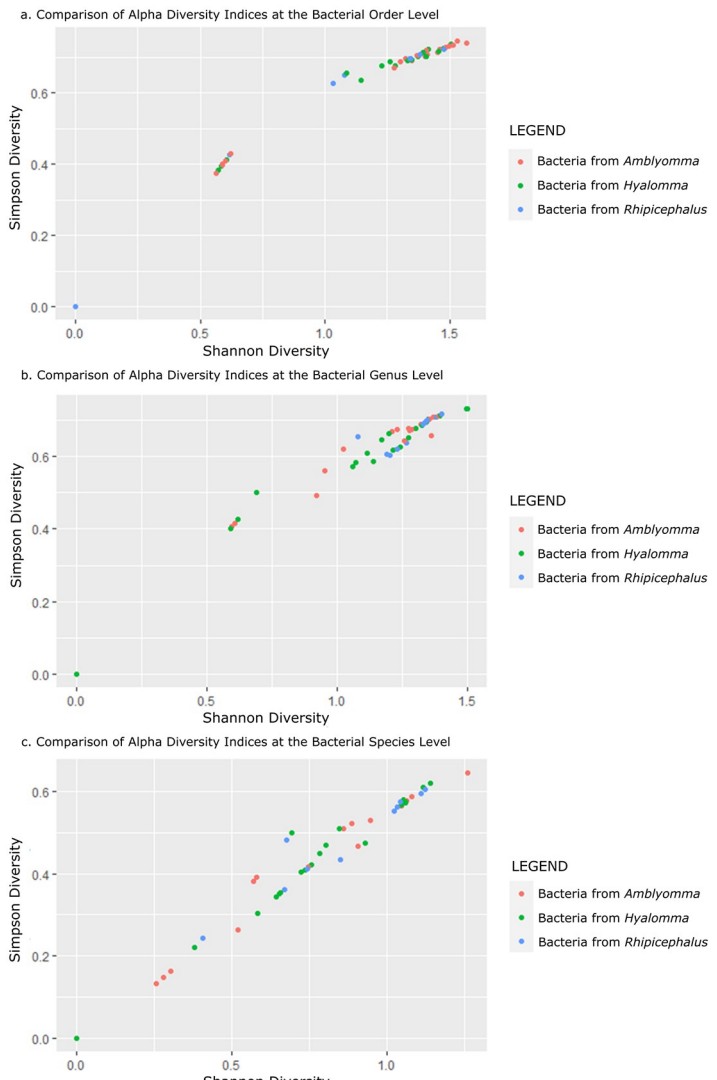

**Fig 3. Comparative scatter plots of Alpha diversity indices of Operational Taxonomic Units at the bacterial order level (a), bacterial genus level (b), and bacterial species level (c).** The three colors represent the tick genus from which they were identified. Note the reduction in richness (Shannon diversity) and an increase in evenness (Simpson diversity) the lower the taxonomic identification level. There is a marked drop in Simpson diversity between the bacterial genus and bacterial species level which coincides with the loss of the Rickettsia genus which could not be speciated.

The PERMANOVA results showed the varying contribution of each of the OTUs at the isolated, individual, and group levels at each of the bacterial taxonomic levels (Table 3). These variances were however not statistically significant ($p$ values > 0.05).

## Taxonomic identification of bacteria and their abundances

Taxonomic read classification using QIIME2 revealed that the highest number of bacterial reads in ticks were from the order Pseudomonadales (46.52%), followed by Rickettsiales (27.53%), Coxiellales (14.57%), Staphylococcales (5.22%), Enterobacterales (4.79%),

**Table 3. PERMANOVA results for beta diversity between tick genera at different bacterial taxonomic classification levels.**

| Bacterial Taxonomic Levels | $R^2$ (Model) | $R^2$ (Conditional on Terms) | $R^2$ (Marginal) | $p$ Value |
|---|---|---|---|---|
| Order | 0.0464647 | 0.9535353 | 1.00 | 0.297 |
| Genus | 0.065051 | 0.934949 | 1.00 | 0.106 |
| Species | 0.06127109 | 0.93872891 | 1.00 | 0.166 |

The results show the contribution of each identified Operational Taxonomic Unit (OTU) minus the effect of other OTUs (model), the contribution of each OTU while holding all other OTUs constant, and the contribution of each OTU while considering the effect of variances in other OTUs.

Propionibacteriales (0.54%), Entomoplasmatales (0.36%), Corynebacteriales (0.25%) and finally Micrococcales (0.23%) (Fig 4a).

Kruskal-Wallis test results showed that there were significant differences (P < 0.0001) among bacterial order abundancies. However, Dunn's multiple comparison tests showed that there was no significant difference in abundancies between Rickettsiales vs Coxiellales, Pseudomonadales vs Rickettsiales, Coxiellales vs Enterobacterales, Coxiellales vs Staphylococcales and Enterobacterales vs Staphylococcales. Median reads across pools for the top three orders as follows: Pseudomonadales 410 (IQR 353–495), Rickettsiales 252 (IQR 172–318), and Coxiellales 146 (IQR 108–187).

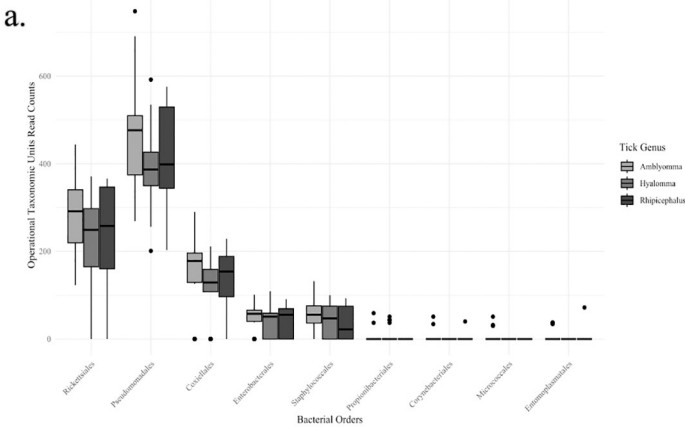

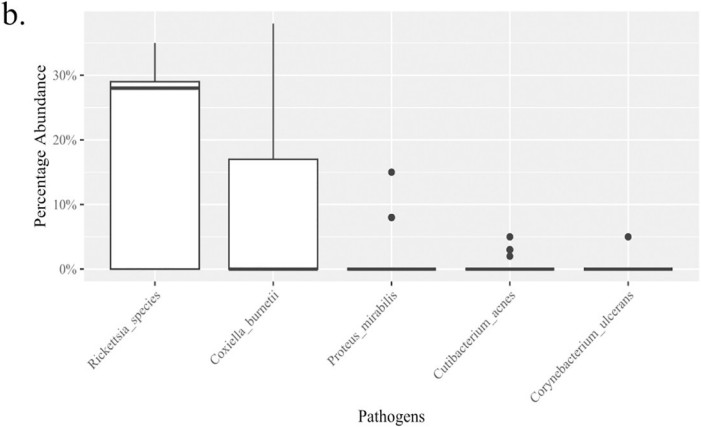

**Fig 4. Abundances at the order level (a) and pathogenic bacteria at the genus and species levels (b).** Significance between the bacterial orders was determined using the Kruskal-Wallis test and yielded a *p*-value of < 0.0001.

**Table 4. Distribution of OTUs that were classified to the bacterial species level per tick genera.**

| | *P. fluorescens* | *C. burnetii* | *P. mirabilis* | *S. agnetis* | *C. acnes* | *C. ulcerans* |
|---|---|---|---|---|---|---|
| *Amblyomma* spp. | 8691 | 1120 | 775 | 508 | 73 | 0 |
| *Hyalomma* spp. | 9831 | 1769 | 1048 | 1174 | 41 | 42 |
| *Rhipicephalus* spp. | 4537 | 1082 | 634 | 604 | 40 | 0 |
| Totals | 23059 | 3971 | 2457 | 2286 | 154 | 42 |

The numbers of Operational Taxonomic Units (OTUs) contributed by each tick genera that were classified (≥60% confidence) to the bacterial species level. *C. ulcerans* was only identified from the *Hyalomma* genus which consistently had the highest count of 5 out of the 6 species identified.

A comparison of the distribution of OTU reads at the order level showed that Pseudomonadales had the highest relative abundance closely followed by Rickettsiales, while the rest had a relative abundance of less than 20% (Fig 4a). For pathogenic and potentially pathogenic bacteria, the Rickettsia and Coxiella genera had the highest abundances (Fig 4b).

Six (6) bacterial OTUs were (≥60%) identified to species level, these are *C. burnetii*, *Pseudomonas fluorescens*, *P. mirabilis*, *Staphylococcus agnetis*, *C. ulcerans*, and *C. acnes* (Table 4). Cytochrome Oxidase I sequence data improved the resolution in the identification of bacteria that use oxidative phosphorylation such as *Coxiella* and *Rickettsia*. These were mostly Rickettsia which could only be identified upto a genus level using 16S rRNA. 38 OTUs were classified into *Rickettsia* species, 16 OTUs were classified into *Rickettsia* subspecies and strains, and 7 OTUs were classified into *C. burnetii* subspecies and strains.

Consideration of bacterial sequences amplified using both 16S rRNA and *COI* approaches allowed for the identification of several pathogenic and potentially pathogenic bacterial species. These were *C. burnetti*, *P.s mirabilis*, *C. acnes*, and *C. ulcerans*. Isiolo had a higher prevalence of pathogenic bacteria, particularly *R. conorii*, while Kwale's leading pathogen was *C. burnetii*. Comparative analysis between tick vectors and potential pathogens was used to determine the distribution of the potential pathogens in ticks (Fig 5). This further revealed that *H. truncatum* and *Rh. boophilus microplus* carried the most unique pathogens in Isiolo and Kwale, respectively.

Additionally the calculation of the Minimum Infection Rates (MIR) per 100 ticks, the distribution of these potential pathogens according to tick hosts showed that ticks collected from

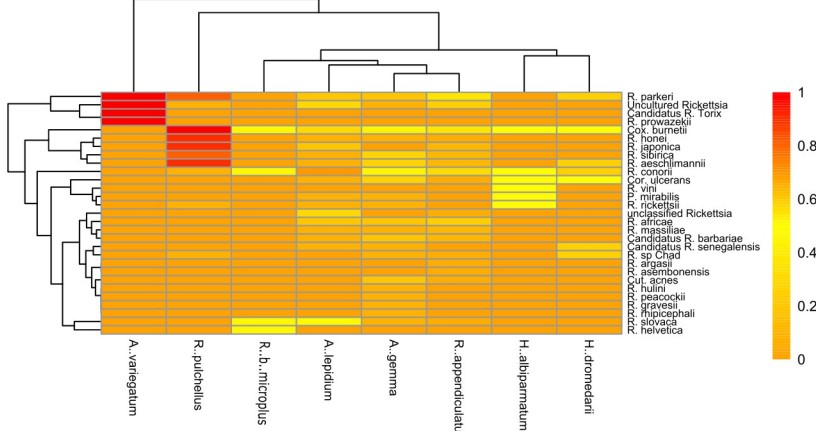

**Fig 5. Heatmap showing the relative abundance of different bacterial pathogens identified.** These include *Rickettsia* species identified using Cytochrome C Oxidase subunit 1 in tick species sampled.

**Table 5. The Minimum Infection Rates (MIR) per 100 ticks collected from different vertebrate hosts.**

| Host | Rickettsia (g) | *Coxiella burnetii* | *Proteus mirabilis* | *Cutibacterium acnes* | *Corynebacterium ulcerans* |
|------|----------------|---------------------|---------------------|-----------------------|----------------------------|
| **Cattle** | 10 | 8 | 1 | 1 | 0 |
| **Sheep** | 16 | 4 | 0 | 3 | 0 |
| **Goats** | 13 | 6 | 2 | 1 | 1 |
| **Camels** | 11 | 2 | 0 | 0 | 2 |

The Ticks collected from cattle had the highest prevalence of all but one potentially pathogenic bacteria. Rickettsia was highly prevalent in ticks from cattle followed closely with ticks from camels. Ticks collected from cattle, sheep, and goats showed the presence of *C. acnes* which might indicate that humans were an alternative source of blood.

cattle had the highest prevalence of potential pathogens and were only surpassed by camels (Table 5).

*COI* sequence analysis on the BugSeq allowed for the speciation of the Rickettsia genus down to the species level. The platform identified 25 *Rickettsia* species of which 11 are known human pathogens: *R. parkeri, R. rickettsii, R. conorii, R. honei, R. slovaca, R. japonica, R. sp Chad, R. aeschlimannii, R. helvetica, R. prowazekii, R. africae, R. vini,* and *R. massiliae. Rickettsia prowazekii* was the only Typhus Group (TG) *Rickettsia* that was identified. The pathogen was identified from *A. variegatum* ticks collected from Kwale County.

## Discussion

Ticks are critical in both the biological and mechanical transmission of many pathogens especially when coupled with their ability to inhabit a wide variety of climates. Since ticks feed on multiple hosts throughout their lifecycles, it has become important under the One Health concept [3,32] to do constant surveillance of pathogens they carry to foster public health preparedness. This study sought to investigate the microbiomes of ticks and by extension the types of human pathogenic bacteria that are present in ticks collected from Kenya.

The number of ticks sampled in the present study was very skewed in favor of Isiolo (n = 2,858) against Kwale County (n = 60). The movement of livestock in Isiolo is higher at both inter and intra-county levels [33] as compared to Kwale County which may explain the higher number of tick infestations in Isiolo. Additionally, this disparity in the collection may be because two of the sites from Isiolo were a livestock market and a slaughterhouse, which would receive a disproportionate number of livestock and by extension ticks.

Previous entomological studies show that *A. variegatum* ticks prefer to feed on horses, cattle, sheep, dogs, and humans as hosts [34]. *Hyalomma* species are known to preferentially feed on rabbits, hares, rodents, and birds while they are nymphs but switch to mostly ruminants once they become adults [35] whereas *Rh. boophilus microplus* has been shown to preferentially feed on cattle [36]. The presence of *C. acnes*, a bacterium that is commonly associated with human skin [37], in the ticks sampled in this study shows potential feeding on human hosts. *Amblyomma* ticks had the highest prevalence of *C. acnes* which can be explained by their higher preference to feed on humans compared to the other species.

The alpha diversity test showed that richness and evenness varied at different taxonomic levels. However, the variability was not statistically significant. This would suggest a similar distribution of the bacterial communities among the different tick genera. Although the bacteriome in ticks is influenced by host, environmental, and trans-ovarian factors [28], commensal or symbiotic relationships also drive the presence of some of these bacteria since they are useful or important for the survival of the ticks [38,39]. These relationships between the ticks

and endosymbiotic bacteria, a good number of which belong to the *Rickettsia* and *Coxiella* genera [40], might explain why the overall bacteriome of different tick species collected from the two different regions is similar to a greater extent as indicated by the alpha diversity indices. This is probably because the presence of some of these endosymbionts increases the general survivability and fitness of the tick [38]. The beta diversity test also indicated that the difference in bacterial abundances between the different genera was not significant. In addition to this, the bacterial groups especially at the order level were somewhat other. Thus, bacterial infestation may be influenced by external factors other than the type of tick. These factors may include the type of host, local environmental conditions such as temperature [41], or transmission between consecutive generations [42].

The present study identified bacteria belonging to the classes of Gammaproteobacteria (Pseudomonadales and Coxiellales), and Alphaproteobacteria (Rickettsiales) as the most abundant bacterial orders sampled in Isiolo and Kwale. The Pseudomonadales identified in this study include *P. fluorescens*, which is a ubiquitous bacterium in the environment that has recently been implicated as a potential pathogen [43]. The majority of Rickettsiales identified in this study belonged to the genus Rickettsia. Prior studies screening ticks have demonstrated the abundant presence of Rickettsiae spp. in Kenya [40–42]. Similarly, Rickettsial diseases such as tick typhus and African tick bite fever are well established in Kenya. This includes a recent fatal case of *R. conorii* infection in an American traveller in Kenya [44] and an *R. felis* infection that afflicted six people in the Northeastern region of Kenya [45]. Rickettsial diseases have persisted in the region for over fifty years; however, there are still knowledge gaps in surveillance and reporting that need to be filled [46].

The most notable member of the Coxiellales identified in this study was *C. burnetii*. These pathogenic intracellular bacteria are responsible for Q Fever, a disease commonly identified in people working close to farm animals or in pastoral environments [47]. The high prevalence of this pathogen amongst ticks sampled from animals in the present study portends a risk to public health. A previous study in Kenya has linked *C. burnetii* transmission in humans to ticks [48]. Camels have been identified as playing a bigger role in the transmission of Q-fever compared to other ruminants [35]. In the present study, ticks collected from camels had an MIR of 2 per 100 regarding *C. burnetii*. Whereas a recent study did not find a significant association between the high prevalence of *C. burnetii* in livestock and their owners [49], children have been shown to have a higher predisposition to infection as this pathogen can be transmitted through breast milk [50]. Comparative analysis of results from this and previous studies has highlighted that there are fluctuations in the detected prevalences of *C. burnetii*. On the lower end, one failed to detect it [51] and another reported a prevalence of 5.5% [52]. On the higher end, a study reported a prevalence of 45.7% [53]. This shows that more research is required to ascertain what other factors may be affecting the prevalence of *C. burnetii* in ticks. This could be, seasonal variations, county, hosts, sampling strategy, methods, and unknown outbreaks. This study included marketplaces and slaughterhouses as part of the sample sites as compared to the others that focused solely on farms [52,54]. Analysis of tick species and bacterial pathogens indicates that all *Rh. pulchellus* pools had *C. burnetii* and four Rickettsia species (*R. honei, R. japonica, R. sibrica,* and *R. aeshlimannii*). This contrasts with an earlier study that reported a 25% prevalence of *C. burnetii* within *Rhipicephalus* tick species [54], this may however be explained by the super-pools of ticks tested which had seventy-nine ticks. This means that testing of more ticks collected at different seasons (longitudinal study) and not oversampling individual animals may be required to ascertain other extraneous variables that may affect the prevalence of *C. burnetii*. *Rickettsia prowazekii* was the only Typhus Group (TG) rickettsia detected. It was localized to *A. variegatum* species of ticks collected from Kwale County. The Typhus Group rickettsiae are commonly associated with fleas and body lice [55]. *Rickettsia*

*prowazekii* infection (epidemic typhus) has a reported fatality rate of 15% that goes even higher in regions of high poverty and low medical access and support [56] such as those in Low- and Middle-Income countries, including Kenya.

This study was unable to detect the presence of any *Anaplasma*, *Theileria*, and *Borrelia* species detected in one of the previous studies [51], conducted on ticks collected from wildlife. This may be due to the ticks in the wildlife study having a wider range of vertebrate hosts thus increasing the pool of potential reservoirs for the pathogens.

Another contrast was seen with *P. mirabilis*, which was detected in relatively lower levels of MIR of 1 per 100 than in a previous study [53]. This difference may be due to the tick surface sterilization step in the current study, which removed external bacterial contaminants as *P. mirabilis* is widely found in soil and water.

This study has advanced the understanding of tick-associated bacteria, especially with the combined use of 16S rRNA and *COI* sequence data to enhance the identification resolution for pathogenic bacteria to the species and strain levels for *Rickettsia* and *Coxiella* [57]. This sets the stage for further characterization of cultivatable bacteria such as *P. mirabilis*, *C. acnes*, and *C. ulcerans*.

Our study, while illuminating, is not without limitations. Firstly, our sampling method, though rigorous, may not fully capture the entire tick population due to the potential localization of ticks in specific microhabitats. Moreover, our reliance on molecular techniques like 16S rRNA sequencing, while highly sensitive, carries inherent biases and risks of contamination. The lack of longitudinal data restricts our understanding of the temporal dynamics of tick-borne pathogens, that may be in flux depending on season. Additionally, our study's focus on Isiolo and Kwale counties, while providing unique insights into these regions, limits the direct extrapolation of findings to areas with distinct environmental and socioeconomic contexts. Finally, this study investigated only ticks and did not collect blood samples from their vertebrate hosts for comparative analysis. This therefore limited the inferences that could be made based on host-vector interactions and their influences on tick microbiome. The lack of standardized reporting of tick-borne pathogens as reported before [58], makes it challenging to provide useful comparisons between studies. Future research should address these limitations, potentially incorporating longitudinal studies across broader geographical regions. Incorporating vertebrate and tick blood meal analysis into future research holds the potential to illuminate the effect of the host on the microbiome of ticks. A standardized format of sampling, pooling, and prevalence reporting needs to be adopted to improve the comparability and synergy of studies. The impact of climate and seasonal variations during sampling should also be taken into consideration for a comprehensive understanding of tick-borne bacterial diseases in Kenya.

## Conclusion

In the current study, both intracellular and extracellular pathogenic bacteria known to cause human diseases were identified in ticks. Of interest is the high prevalence of *C. burnetii* the causative agent of Q-fever and various pathogenic Rickettsia species identified in both Isiolo and Kwale, including *R. prowazekii* identified only in Kwale County. The lack of significant variation in microbiomes among the different tick species infers that ticks in the sampled areas share a core microbiome. This study has shown that analysis of full-length *COI* has the potential to be used to simultaneously identify bacteria that use oxidative phosphorylation for energy needs and their tick vectors to species level.

This study has highlighted the high prevalence of various pathogenic *Rickettsia* species and *C. burnetti* in these regions to increase public health preparedness. The presence of *C. acnes* a

bacterium commonly associated with human skin may be indicative of potential human feeding and exposure to the other pathogens. In light of this, there is a need to advise clinicians in these "at-risk" areas to include rickettsial diseases and query fever as part of their differential diagnosis. The viability of these bacteria and the competency of ticks to transmit them should be assessed in future studies.

## Supporting information

**S1 Fig. Percentage stacked bar charts showing the abundances at the bacterial order level (a), genus level (b), and species level (c) per sample.** Samples with the' ISL' prefix are from Isiolo while those with the 'KWL' prefix are from Kwale.
(TIF)

**S1 Table. Distribution of ticks according to species and site.**
(PDF)

## Acknowledgments

We thank Francis Ngere, Nicholas Odemba, David Oullo, Charles Waga, Richard Ochieng, Daniel Ngonga, and Vitalice Opondo for their expert contribution to tick sampling and identification. We also thank Samuel Owaka for his contribution to generating the map of sampling sites.

The material has been reviewed by the Walter Reed Army Institute of Research. There is no objection to its presentation and/or publication. The opinions or assertions contained herein are the private views of the authors and are not to be construed as official or as reflecting the true views of the Department of the Army or the Department of Defence.

## Author Contributions

**Conceptualization:** Bryson Brian Kimemia, Lillian Musila, Solomon Langat, Samoel Khamadi, Fredrick Eyase.

**Data curation:** Bryson Brian Kimemia, Solomon Langat, Santos Yalwala.

**Formal analysis:** Bryson Brian Kimemia, Lillian Musila, Solomon Langat, Santos Yalwala, Samoel Khamadi, Fredrick Eyase.

**Funding acquisition:** Stephanie Cinkovich, David Abuom, Jaree Johnson, Eric Garges, Elly Ojwang, Fredrick Eyase.

**Investigation:** Bryson Brian Kimemia, Solomon Langat, Erick Odoyo, Stephanie Cinkovich, Santos Yalwala, Samoel Khamadi, Elly Ojwang.

**Methodology:** Bryson Brian Kimemia, Lillian Musila, Solomon Langat, Erick Odoyo, Santos Yalwala, Fredrick Eyase.

**Project administration:** Erick Odoyo, Stephanie Cinkovich, David Abuom, Santos Yalwala, Jaree Johnson, Eric Garges, Elly Ojwang, Fredrick Eyase.

**Resources:** Lillian Musila, Stephanie Cinkovich, David Abuom, Santos Yalwala, Jaree Johnson, Eric Garges, Elly Ojwang, Fredrick Eyase.

**Software:** Bryson Brian Kimemia, Solomon Langat.

**Supervision:** Lillian Musila, David Abuom, Samoel Khamadi, Jaree Johnson, Eric Garges, Elly Ojwang, Fredrick Eyase.

**Validation:** Bryson Brian Kimemia, Lillian Musila, Samoel Khamadi, Eric Garges, Fredrick Eyase.

**Visualization:** Bryson Brian Kimemia, Erick Odoyo, Fredrick Eyase.

**Writing – original draft:** Bryson Brian Kimemia, Lillian Musila, Solomon Langat, Erick Odoyo, Stephanie Cinkovich, David Abuom, Santos Yalwala, Samoel Khamadi, Jaree Johnson, Eric Garges, Elly Ojwang, Fredrick Eyase.

**Writing – review & editing:** Bryson Brian Kimemia, Lillian Musila, Solomon Langat, Erick Odoyo, Stephanie Cinkovich, David Abuom, Santos Yalwala, Samoel Khamadi, Jaree Johnson, Eric Garges, Elly Ojwang, Fredrick Eyase.

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
