## [Decision Letter · Decision Letter 0]

22 Jan 2024

PONE-D-23-42435Detection of pathogenic bacteria in ticks from Isiolo and Kwale counties of Kenya using metagenomics.PLOS ONE

Dear Dr. Kimemia,

Thank you for submitting your manuscript to PLOS ONE. After careful consideration, we feel that it has merit but does not fully meet PLOS ONE’s publication criteria as it currently stands. Therefore, we invite you to submit a revised version of the manuscript that addresses the points raised during the review process.

**ACADEMIC EDITOR: **The manuscript needs a revision according to the reviewers' notes and suggestions.

We look forward to receiving your revised manuscript.

Kind regards,

Shawky M Aboelhadid, PhD

Academic Editor

PLOS ONE

“We thank Francis Ngere, Nicholas Odemba, David Oullo, Charles Waga, Richard Ochieng, Daniel Ngonga and Vitalice Opondo for their expert contribution in ticks sampling and identification.

This work was funded by the Armed Forces Health Surveillance Branch (AFHSB) and its Global Emerging Infections Surveillance (GEIS) Section, FY2022 ProMIS ID: P0116_22_KY and FY2023 ProMIS ID P0094_23_KY. The material has been reviewed by the Walter Reed Army Institute of Research. There is no objection to its presentation and/or publication. The opinions or assertions contained herein are the private views of the authors and are not to be construed as official or as reflecting the true views of the Department of the Army or the Department of Defence.”

6. We note that Figure 1 in your submission contain [map/satellite] images which may be copyrighted. All PLOS content is published under the Creative Commons Attribution License (CC BY 4.0), which means that the manuscript, images, and Supporting Information files will be freely available online, and any third party is permitted to access, download, copy, distribute, and use these materials in any way, even commercially, with proper attribution. For these reasons, we cannot publish previously copyrighted maps or satellite images created using proprietary data, such as Google software (Google Maps, Street View, and Earth). For more information, see our copyright guidelines: http://journals.plos.org/plosone/s/licenses-and-copyright.

Reviewers' comments:

Reviewer's Responses to Questions

**Comments to the Author**

1. Is the manuscript technically sound, and do the data support the conclusions?

Reviewer #1: Yes

Reviewer #2: Yes

Reviewer #3: Yes

2. Has the statistical analysis been performed appropriately and rigorously? 

Reviewer #1: N/A

Reviewer #2: Yes

Reviewer #3: Yes

3. Have the authors made all data underlying the findings in their manuscript fully available?

Reviewer #1: Yes

Reviewer #2: Yes

Reviewer #3: No

4. Is the manuscript presented in an intelligible fashion and written in standard English?

Reviewer #1: No

Reviewer #2: Yes

Reviewer #3: Yes

5. Review Comments to the Author

Reviewer #1: Ticks were collected from cattle, sheep, goats, and camels, therefore, these ticks must have sucked blood. The microorganisms should be affected by the blood from different hosts. When ticks were grouped, the host they parasitized should be considered. For example, Rickettsia prowazekii are commonly asociated with fleas and body lice, and it identified in ticks may be from blood. Therefore, these limitation should be added in the disscussion.

When detecting the microorganisms, the ticks were pooled in groups of 1 to 8 individuals. Therefore, it is unreasonable to calculate the prevalence of microorganisms, and even the prevalence compared to previous studies was also unreasonable.

Table 4 was not shown in the text.

1. The MS should be reviewed by a native english speaker.

2. Species names: with full genus name at first mention, later with abbreviated genus name throughout the text (exception: at the beginning of a sentence). Species names and genus names should be italicized. Please check throughout the text.

3. "while the second was to confirm the morphological tick species identification", hence, can the COI gene of microorganisms also be amplified with the primers provided in the MS?

4. "COI" and "p" value should be italicized.

5. "Spotted Fever Group Rickettsias", replaced by "Spotted Fever Group of Rickettsiae".

6. table 4 was not shown in text, how to deternine the prevalence?

7. line 135, add as before per.

8. line 290, Pseudomonadales 410, please show IQR.

9. line 329, add "," after Rickettsia slovaca.

10. the section "Confirmation of Vector Identity using Cytochrome Oxidase Subunit I Sequences" should be shown in section "Tick Collection".

11. line 367, the endosymbiotic bacteria was not identified in this study, how to "explain why the overall bacteriome of different tick species collected from the two different regions is similar to a greater extent as indicated by the alpha diversity indices"?

Reviewer #2: The study highlights the occurrence of pathogenic bacteria of public health importance in tick species. The findings show the need for continuous surveillance efforts to create effective control and preventive measures.

I have a few suggestions to improve the manuscript.

Comments

L112-113: Did the authors consider pooling the ticks based on the animal host? If yes, please include it in the “Sampling and Pooling” section.

L139-140: The word “were” was omitted. The sentence should read…. reverse primer pair were used whereas…

L145: Remove the “min” after the “denaturation for 30 sec”.

L148: Remove the “min” after the “95ºC denaturation for 30 sec”.

Results:

For the tick collection section, the authors should consider first talking about the tick collection at the various sampled areas, the tick species identified and then zoom in to the pools.

Which of the livestock sampled were mostly infested by tick species? The authors can consider including the number of ticks collected from each livestock (cattle, sheep, goat, camel) in the results.

L202-206: The authors should consider revising the statements made for clarity.

L249-252: Kindly read the sentence and make the necessary corrections.

Table 2 captioned “Distribution of OTUs that were classified to the bacterial species level per Tick Genera” and Table 3 “The numbers of Operational Taxonomic Units (OTUs) contributed by each tick genera that were classified (≥60% confidence) to the bacterial species level” are practically the same. Remove the Table 2.

L402: The authors mentioned Table 4, but no Table 4 has been presented in the results.

There are two different captions of Figure 1 in the manuscript. I believe the second Figure 1 mentioned in the discussion (L432-434) is Figure 5. Please confirm and make the change.

Reviewer #3: Line 71: correct to “….reported spotted fever group Rickettsia in the United……..”

Line 97: “The study was also submitted for ……………..

Line 108: 15 mL

Line 121: 1 mL

Line 127: 4 m/s

Line 128: 1000 µL, also in line 130, correct to 200 µL

Line 136: 16S rRNA

Line 150: is 0.8% agarose gel not too low?

Line 201-203: The sentence is confusing. The way it began, it sounds like the total number collected from Isiolo was 2918 which is not so. 2918 was the total number collected from both Isiolo and Kwale. The sentence needs to be revised for clarity by stating explicitly, the total overall number collected and the total collected each from Isiolo and Kwale.

Line 205: it must be exactly and not approximately.

Line 207: The authors have mentioned that the collected tick’s falls into 10 species in total but in line 205, it was mentioned that the 372 tick pools represented 8 ticks. This is conflicting and needs to be clarified.

Line 215-217: this line of text is completely distractive considering its location. Preferably, this should come immediately after line 212.

Line 346-347: skewed in term of what? Numbers? If so, clearly and expressly state so and avoid ambiguous terms that are meaningless without been qualified.

Line 349-350”: The use of the word “respectively” in this sentence is meaningless as it has nothing to qualify here. Take a look and revise

Introduction

In the current form, the introduction is too generalist without focus and the background information is too scanty that attempts to provide good background. The authors need to rejig the introduction by making it more focused.

Methodology

They is need for the authors to provide details of the geography of the sampling locations including details of the average temperature, humidity and seasons. I understand that part of this information has been provided in the introduction; however, this can be removed and transferred to the methodology. Usually, the first part of the methodology deals with the description of the “study area”. In the study area, the authors should state specifically, the months and year when tick sampling was undertaken. In other words, the duration of sampling is necessary.

Discussion

Some of the results were embedded within the discussion section which is a misnormal and needs to be revised. For instance, table 2 and figure 1 was presented in between the text for the discussion. Take another look at this and address my concerns.

Results

Confirmation of the vector identity should be the first part of the results after stating the total number of ticks collected across all the study locations and the number of ticks. It is abnormal to state all the pathogens detected in the various tick species before providing details of how the tick vector identity was confirmed.

General comments

-Numerous grammatical and editorial errors that makes it difficult to grasp the meaning of some of the sentences.

- In the discussion, I will suggest that the authors devise a more robust way to discuss their findings. For instance, discuss the tick prevalence and distribution first, then tick borne alpha-proteobacteria that are potentially pathogenic, zoonotic and non cultivable before discussing the findings from gram negative bacteria that can be cultured on media.

6. PLOS authors have the option to publish the peer review history of their article (what does this mean?). If published, this will include your full peer review and any attached files.

Reviewer #1: No

Reviewer #2: No

Reviewer #3: **Yes: **THANKGOD EMMANUEL ONYICHE

---

## [Author Response · Author response to Decision Letter 0]

27 Feb 2024

Responses to Editorial Comments

Comment 1: Please ensure that your manuscript meets PLOS ONE’s style requirements, including those for file naming. The PLOS ONE style templates can be found at

Response 1: The manuscript has been formatted according to PLOS ONE’s styling requirements.

Comment 2: In your Methods section, please provide additional information regarding the permits you obtained for the work. Please ensure you have included the full name of the authority that approved the field site access and, if no permits were required, a brief statement explaining why.

Response 2: Ethical approval was sought from Kenya Medical Research Institute’s (KEMRI) Scientific and Ethics Review Unit (SERU) and permission to carry out the study in the country was provided by the National Council for Science, Technology and Innovation (NACOSTI) as outlined in the Ethical Approval section.

Comment 3: We suggest you thoroughly copyedit your manuscript for language usage, spelling, and grammar. If you do not know anyone who can help you do this, you may wish to consider employing a professional scientific editing service.

Response 3: The manuscript was reviewed, edited, and any grammatical and editorial errors corrected by the authors as evidenced by the attached tracked corrected copy of the manuscript.

Comment 4: Thank you for stating the following in the Acknowledgments Section of your manuscript:

“We thank Francis Ngere, Nicholas Odemba, David Oullo, Charles Waga, Richard Ochieng, Daniel Ngonga and Vitalice Opondo for their expert contribution in ticks sampling and identification.

This work was funded by the Armed Forces Health Surveillance Branch (AFHSB) and its Global Emerging Infections Surveillance (GEIS) Section, FY2022 ProMIS ID: P0116_22_KY and FY2023 ProMIS ID P0094_23_KY. The material has been reviewed by the Walter Reed Army Institute of Research. There is no objection to its presentation and/or publication. The opinions or assertions contained herein are the private views of the authors and are not to be construed as official or as reflecting the true views of the Department of the Army or the Department of Defence.”

Response 4: The funding statement was removed from the Manuscript text and the authors would like to update the Funding Statement in the submission to read “This work was funded by the Armed Forces Health Surveillance Branch (AFHSB) and its Global Emerging Infections Surveillance (GEIS) Section, FY2022 ProMIS ID: P0116_22_KY and FY2023 ProMIS ID P0094_23_KY.”

Comment 5: When completing the data availability statement of the submission form, you indicated that you will make your data available on acceptance. We strongly recommend all authors decide on a data sharing plan before acceptance, as the process can be lengthy and hold up publication timelines. Please note that, though access restrictions are acceptable now, your entire data will need to be made freely accessible if your manuscript is accepted for publication. This policy applies to all data except where public deposition would breach compliance with the protocol approved by your research ethics board. If you are unable to adhere to our open data policy, please kindly revise your statement to explain your reasoning and we will seek the editor’s input on an exemption. Please be assured that, once you have provided your new statement, the assessment of your exemption will not hold up the peer review process.

Response 5: The sequence data uploaded to NCBI was not available by the time of manuscript submission. It was successfully reviewed and can be accessed under the BioProject Accession Number PRJNA1053293. The sequence data has been assigned BioSample accession numbers SAMN38855066 to SAMN38855120. The sequence data has been assigned SRA accession numbers (accession numbers - SRS19927750, SRS19927751, SRS19927763, SRS19927772, SRS19927785, SRS19927795, SRS19927801, SRS19927802, SRS19927803, SRS19927804, SRS19927752, SRS19927753, SRS19927754, SRS19927755, SRS19927756, SRS19927757, SRS19927758, SRS19927759, SRS19927760, SRS19927762, SRS19927761, SRS19927764, SRS19927765, SRS19927767, SRS19927766, SRS19927768, SRS19927769, SRS19927770, SRS19927773, SRS19927771, SRS19927774, SRS19927775, SRS19927776, SRS19927777, SRS19927778, SRS19927779, SRS19927780, SRS19927781, SRS19927782, SRS19927784, SRS19927783, SRS19927786, SRS19927787, SRS19927789, SRS19927788, SRS19927790, SRS19927792, SRS19927791, SRS19927793, SRS19927794, SRS19927798, SRS19927799, SRS19927797, SRS19927796, SRS19927800)

Comment 6: We note that Figure 1 in your submission contain [map/satellite] images which may be copyrighted. All PLOS content is published under the Creative Commons Attribution License (CC BY 4.0), which means that the manuscript, images, and Supporting Information files will be freely available online, and any third party is permitted to access, download, copy, distribute, and use these materials in any way, even commercially, with proper attribution. For these reasons, we cannot publish previously copyrighted maps or satellite images created using proprietary data, such as Google software (Google Maps, Street View, and Earth). For more information, see our copyright guidelines: http://journals.plos.org/plosone/s/licenses-and-copyright.

Response 6: The map images were generated using ArcGIS version 10.2.2 for Desktop (Advanced License) by Samuel Owaka using GPS Coordinates collected during the sampling process. Line 124 - 126

Responses to Reviewer Comments

1. Reviewer 1

Comment 1: Ticks were collected from cattle, sheep, goats, and camels, therefore, these ticks must have sucked blood. Microorganisms should be affected by the blood from different hosts. When ticks were grouped, the host they parasitized should be considered. For example, Rickettsia prowazekii are commonly asociated with fleas and body lice, and it identified in ticks may be from blood. Therefore, these limitation should be added in the disscussion.

When detecting the microorganisms, the ticks were pooled in groups of 1 to 8 individuals. Therefore, it is unreasonable to calculate the prevalence of microorganisms, and even the prevalence compared to previous studies was also unreasonable.

Response 1: The Minimum Infection Rate (MIR) was used to represent the minimum number of ticks infected and accounts for the pooling. Line 225 – 228.

Comment 2: Table 4 was not shown in the text.

Response 2: The table was inserted into the Manuscript and updated as Table 5. Line 396 -401.

Comment 3: The MS should be reviewed by a native English speaker.

Response 3: The manuscript was reviewed, edited, and any grammatical and editorial errors corrected by the authors as evidenced by the attached tracked corrected copy of the manuscript.

Comment 4: Species names: with full genus name at first mention, later with abbreviated genus name throughout the text (exception: at the beginning of a sentence). Species names and genus names should be italicized. Please check throughout the text.

Response 4: This was done and updated all throughout the manuscript.

Comment 5: “while the second was to confirm the morphological tick species identification”, hence, can the COI gene of microorganisms also be amplified with the primers provided in the MS?

Response 5: Yes, using these primers the Cytochrome C Oxidase subunit I gene from bacteria that obtain energy from oxidative phosphorylation can be amplified.

Comment 6: “COI” and “p” value should be italicized.

Response 6: This was done and updated all throughout the manuscript.

Comment 7: “Spotted Fever Group Rickettsias”, replaced by “Spotted Fever Group of Rickettsiae”.

Response 7: This was done. Line 74.

Comment 8: table 4 was not shown in text, how to deternine the prevalence?

Response 8: The table was inserted into the Manuscript and updated as Table 5. Line 396 -401. The Minimum Infection Rate (MIR) was used to represent the minimum number of ticks infected and accounts for the pooling. Line 225 – 228.

Comment 9: line 135, add as before per.

Response 9: This was done. Line 159.

Comment 10: line 290, Pseudomonadales 410, please show IQR.

Response 10: This was done through the removal of the statement “interquartile range” that was confusing. Line 351.

Comment 11: line 329, add “,” after Rickettsia slovaca.

Response 11: This was done. Line 407.

Comment 12: the section “Confirmation of Vector Identity using Cytochrome Oxidase Subunit I Sequences” should be shown in section “Tick Collection”.

Response 12: The section was moved to the “Tick Collection” which was renamed “Tick Collection and Identification”. Lines 232 & 266 – 271.

Comment 13: line 367, the endosymbiotic bacteria was not identified in this study, how to “explain why the overall bacteriome of different tick species collected from the two different regions is similar to a greater extent as indicated by the alpha diversity indices”?

Response 13: Endosymbiotic bacteria belong to several genera which include Rickettsia and Coxiella that were identified in this study. Using 16S rRNA data the Rickettsia genus was not speciated and the authors postulate that some of these endosymbionts may be part of the unidentified Rickettsia species. Line 441 – 443.

2. Reviewer 2

Comment 1: L112-113: Did the authors consider pooling the ticks based on the animal host? If yes, please include it in the “Sampling and Pooling” section.

Response 1: Yes, the ticks were also pooled based on the animal host. This has been updated in the “Sampling and Pooling” section. Line 133.

Comment 2: L139-140: The word “were” was omitted. The sentence should read ”reverse primer pair were used whereas”

Response 2: This has been corrected. Line 168.

Comment 3: L145: Remove the “min” after the “denaturation for 30 sec”.

Response 3: This has been corrected. Line 171.

Comment 4: L148: Remove the “min” after the “95oC denaturation for 30 sec”.

Response 4: This has been corrected. Line 174.

Comment 5: Results: For the tick collection section, the authors should consider first talking about the tick collection at the various sampled areas, the tick species identified and then zoom in to the pools.

Response 5: The section has been reworked to incorporate the reviewer’s recommendations. Line 232 - 249

Comment 6: Which of the livestock sampled were mostly infested by tick species? The authors can consider including the number of ticks collected from each livestock (cattle, sheep, goat, camel) in the results.

Response 6: This information has been included in the “Tick Collection and Identification” section as Table 1. Lines 249 - 250

Comment 7: L202-206: The authors should consider revising the statements made for clarity.

Response 7: The statements have been reworked to improve clarity. Line 244 – 247.

Comment 8: L249-252: Kindly read the sentence and make the necessary corrections.

Response 8: This has been corrected to show the impact of overrepresented OTUs more adequately on the tick microbiome diversity. Line 302 - 306.

Comment 9: Table 2 captioned “Distribution of OTUs that were classified to the bacterial species level per Tick Genera” and Table 3 “The numbers of Operational Taxonomic Units (OTUs) contributed by each tick genera that were classified (>60% confidence) to the bacterial species level” are practically the same. Remove the Table 2.

Response 9: Table 2 (now Table 3) was about Beta diversity measured using PERMANOVA while Table 3 (now Table 4) was about the distribution of classified reads.

Comment 10: L402: T

---

## [Decision Letter · Decision Letter 1]

12 Mar 2024

PONE-D-23-42435R1Detection of pathogenic bacteria in ticks from Isiolo and Kwale counties of Kenya using metagenomics.PLOS ONE

Dear Dr. Kimemia,

Thank you for submitting your manuscript to PLOS ONE. After careful consideration, we feel that it has merit but does not fully meet PLOS ONE’s publication criteria as it currently stands. Therefore, we invite you to submit a revised version of the manuscript that addresses the points raised during the review process.

**ACADEMIC EDITOR: **The authors should revise the manuscript according to reviewers comments. The comments of a reviewer are not replied to by the authors. 

We look forward to receiving your revised manuscript.

Kind regards,

Shawky M Aboelhadid, PhD

Academic Editor

PLOS ONE

Reviewers' comments:

Reviewer's Responses to Questions

**Comments to the Author**

1. If the authors have adequately addressed your comments raised in a previous round of review and you feel that this manuscript is now acceptable for publication, you may indicate that here to bypass the “Comments to the Author” section, enter your conflict of interest statement in the “Confidential to Editor” section, and submit your "Accept" recommendation.

Reviewer #2: All comments have been addressed

Reviewer #3: (No Response)

2. Is the manuscript technically sound, and do the data support the conclusions?

Reviewer #2: Yes

Reviewer #3: No

3. Has the statistical analysis been performed appropriately and rigorously? 

Reviewer #2: Yes

Reviewer #3: Yes

4. Have the authors made all data underlying the findings in their manuscript fully available?

Reviewer #2: Yes

Reviewer #3: No

5. Is the manuscript presented in an intelligible fashion and written in standard English?

Reviewer #2: Yes

Reviewer #3: Yes

6. Review Comments to the Author

Reviewer #2: (No Response)

Reviewer #3: The author failed to address all my comments made in the earlier revision. I will recommend that they address my earlier comments or make satisfactory rebuttal to all my comments

7. PLOS authors have the option to publish the peer review history of their article (what does this mean?). If published, this will include your full peer review and any attached files.

Reviewer #2: No

Reviewer #3: **Yes: **ThankGod Emmanuel Onyiche

---

## [Author Response · Author response to Decision Letter 1]

6 Apr 2024

The responses are contained in the "Response to Reviewers Letter R2"

---

## [Editor Report · Decision Letter 2]

10 Apr 2024

Detection of pathogenic bacteria in ticks from Isiolo and Kwale counties of Kenya using metagenomics.

PONE-D-23-42435R2

Dear Dr. Kimemia,

We’re pleased to inform you that your manuscript has been judged scientifically suitable for publication and will be formally accepted for publication once it meets all outstanding technical requirements.

Kind regards,

Shawky M Aboelhadid, PhD

Academic Editor

PLOS ONE